# Multirate Audio-Integrated Feedback Active Noise Control Systems Using Decimated-Band Adaptive Filters for Reducing Narrowband Noises [note 1]

**DOI:** 10.3390/s20226693

**Published:** 2020-11-23

**Authors:** Antonius Siswanto, Cheng-Yuan Chang, Sen M. Kuo

**Affiliations:** Department of Electrical Engineering, Chung Yuan Christian University, Taoyuan 32023, Taiwan; g10278040@cycu.edu.tw (A.S.); smkuo@cycu.edu.tw (S.M.K.)

**Keywords:** multirate, feedback active noise control, audio-integrated, noise control

## Abstract

Audio-integrated feedback active noise control (AFANC) systems deliver wideband audio signals and cancel low frequency narrowband noises simultaneously. The conventional AFANC system uses single-rate processing with fullband adaptive active noise control (ANC) filter for generating anti-noise signal and fullband audio cancelation filter for audio-interference cancelation. The conventional system requires a high sampling rate for audio processing. Thus, the fullband adaptive filters require long filter lengths, resulting in high computational complexity and impracticality in real-time system. This paper proposes a multirate AFANC system using decimated-band adaptive filters (DAFs) to decrease the required filter lengths. The decimated-band adaptive ANC filter is updated by the proposed decimated filtered-X least mean square (FXLMS) algorithm, and the decimated-band audio cancelation filter can be obtained by the proposed on-line and off-line decimated secondary-path modeling algorithms. The computational complexity can be decreased significantly in the proposed AFANC system with good enough noise reduction and fast convergence speed, which were verified in the analysis and computer simulations. The proposed AFANC system was implemented for an active headrest system, and the real-time performances were tested in real-time experiments.

## 1. Introduction

Excessive noise exposure is harmful to human health and can lead to cardiovascular disorders, cognitive impairment, sleep disturbance, tinnitus, and annoyance [1]. Individual performances are also decreased by high noise exposure [2]. In addition, the economic impact of noise is also significant. The estimated cost of urban road noise in England is £7 to 10 billion annually [3]. Reducing the environmental noise by 5 dB would gain $3.9 billion annual economic benefit in the United States [4]. Thus, noise should be reduced [5] to protect humans from the negative impacts and enhance the soundscape [6,7,8].

Noise can be reduced by passive noise control methods such as using soundproofing enclosures [9] or erecting noise barriers to suppress the noise transmission paths [10,11,12]. Even though the passive control methods could achieve high noise reduction over a broad frequency range, however, the implementations for low frequency noises are bulky and costly due to the long acoustic wavelengths [13]. Alternatively, active noise control methods offer effective low frequency noise reduction by applying additional secondary sources to cancel noise from the original primary source, based on superposition [14]. 

The control structure of active noise control (ANC) is generally classified into feedforward control and feedback control. In feedforward control structure, a reference sensor is required to provide correlating reference noise information for the controller in order to generate the correlating anti-noise sound played by the secondary source. The error sensor picks up the residual noise serving as the error signal for updating the controller. In practical application where the noise source location is unknown, the reference noise cannot be determined. Hence, the feedback control structure can be applied where a reference sensor is not required. Analog or digital based controllers can be used for the feedback control structure.

Analog feedback ANC is based on negative feedback control system, where the performance is influenced by the waterbed effect and the effective bandwidth caused by delay [15]. In order to minimize the delay, the error sensor is required to be placed close to the secondary source. This arrangement is suitable for limited applications such as ANC headphones [16,17,18]. The other approach is to use digital feedback ANC based on the internal model control (IMC) system [19], where the controller acts as the predictor. Hence, a digital feedback ANC system can only reduce predictable noise [14,15]. The digital feedback controller can be made adaptive to have accurate and specific frequency response including effective adjustment of the waterbed effect [20]. This kind of system is called the adaptive feedback ANC system [14] and is well suited to reduce narrowband noise rather than broadband noise [15]. Narrowband noises usually exist in living environments [21,22] such as noises produced by rotating machines [23,24,25].

Audio-integrated feedback active noise control (AFANC) systems are an integration of adaptive feedback ANC systems and audio systems to achieve narrowband noise reduction and deliver wideband audio sound simultaneously. The overall cost is reduced by sharing the audio system components, at which the anti-noise and music sounds are mixed and produced by the same loudspeakers [26]. However, the error signals measured by the microphones contain the desired audio components, which are received as undesired noises, and can be reduced by the AFANC systems. Since audio sounds may contain both narrowband and broadband frequency components, the AFANC system would reduce the narrowband component of the audio, which degrades the music sound quality. On the other hand, the broadband frequency components of the audio become disturbances to the ANC adaptation process, which degrade the noise reduction performance. Therefore, the AFANC system requires intelligent enough algorithms to reduce undesired noises and keep the audio sound at the same time.

The audio-integrated feedforward ANC system was first proposed for use in automobile passenger compartment applications [26]. This conventional system uses the audio-interference cancellation algorithm to remove the audio component signal from the error signal. Therefore, the audio-interference cancellation algorithm combined with the feedforward ANC algorithm can achieve undesired noise reduction and keep the audio simultaneously. The same idea has been extended for AFANC system for several applications such as headsets [27], head-mounted ANC systems, and earphones applications [28,29].

All these AFANC systems operate on a single sampling rate system. Typically, ANC systems are effective for low frequency noise reduction, thus, the required sampling rate is low. However, audio systems require higher sampling rate, typically 8 kHz for speech signal and 44.1 kHz for music signal [27]. If low sampling rate is used for both ANC and audio systems, high frequency audio components will be lost. On the other hand, if a high sampling rate is used, the ANC systems will require longer filter lengths, and be exposed to higher frequency noises, degrading the performances.

Therefore, this paper proposes multirate processing instead of single-rate processing in the conventional AFANC system. The proposed system uses DAF for ANC and audio cancelation filters with short filter lengths. The FXLMS algorithm is used to update the decimated-band ANC filter. Furthermore, the decimated-band audio cancelation filter can be obtained by the proposed on-line and off-line decimated-band secondary-path modeling algorithms. With shorter filter lengths, the computational complexity is significantly reduced in the proposed AFANC system, and good narrowband noise reduction and fast enough convergence are maintained.

The rest of the paper is organized as follows. Section 2 presents the conventional and proposed AFANC systems including the computational reduction and convergence analysis. Section 3 provides the simulations and experiment results. Conclusions are given in Section 4.

## 2. Audio-Integrated Feedback Active Noise Control (AFANC) Systems

This section introduces the methodology of the conventional AFANC system including the conventional off-line secondary-path modeling. Then, the methodologies of the proposed AFANC system and the proposed off-line multirate transfer function modeling are described in detail. The performances of both systems in terms of computational complexity and convergence speed are then analyzed and compared.

### 2.1. Conventional Algorithm Using Fullband Adaptive Filters

The conventional AFANC system employs two adaptive filters, one for ANC, and the other for audio-interference cancelation. The overall system operates on a single sampling rate $f_{s}$, thus the adaptive filters are fullband filters, as illustrated in Figure 1. The adaptive finite impulse response (FIR) fullband ANC filter W(z) generates anti-noise y(n) using the synthesized reference signal x(n) expressed as
(1)$$y\left(n\right)=w^{T}\left(n\right)x\left(n\right),$$
where T denotes transpose operation; n is the time index with sampling rate $f_{s}$; $w\left(n\right)={\left[\begin{array}{cccc} w_{0}\left(n\right) & w_{1}\left(n\right) & \cdots & w_{L-1}\left(n\right) \end{array}\right]}^{T}$ is the weight vector; and $w_{l}\left(n\right)$ is the $l^{\mathrm{th}}$ coefficient of W(z) with filter length L. $x\left(n\right)={\left[\begin{array}{cccc} x\left(n\right) & x\left(n-1\right) & \cdots & x\left(n-L+1\right) \end{array}\right]}^{T}$ is the synthesized reference signal vector.

The audio signal a(n) provided by a digital audio source is summed with the anti-noise signal, resulting in driving signal u(n) given by
(2)u(n)=y(n)+a(n).

After passing through the secondary-path S(z), which includes a digital-to-analog converter (DAC), reconstruction filter, power amplifier, secondary loudspeaker, acoustic-path, error microphone, pre-amplifier, anti-aliasing filter, and analog-to-digital converter (ADC), the system obtains the error signal given by
(3)$$e\left(n\right)=d\left(n\right)-u^{\prime }\left(n\right)=d\left(n\right)-\left[y^{\prime }\left(n\right)+a^{\prime }\left(n\right)\right],$$
where d(n) is the undesired noise signal; $u^{\prime }\left(n\right)$ is the cancelation signal that contains the anti-noise component $y^{\prime }\left(n\right)$; and the audio component is $a^{\prime }\left(n\right)$. The residual noise component $\left(d\left(n\right)-y^{\prime }\left(n\right)\right)$ is necessary for ANC function, but the audio component $a^{\prime }\left(n\right)$ becomes an interference to the adaptation of W(z).

Therefore, the audio cancellation algorithm is necessary to remove the unwanted audio component $a^{\prime }\left(n\right)$ from the error signal e(n) by employing an audio-interference cancelation filter $\hat{S}\left(z\right)$ to estimate $a^{\prime }\left(n\right)$ using the digital audio signal a(n) as the reference signal, given by
(4)$${\hat{a}}^{\prime }\left(n\right)={\hat{s}}^{T}\left(n\right)a\left(n\right),$$
where $\hat{s}\left(n\right)={\left[\begin{array}{cccc} {\hat{s}}_{0}\left(n\right) & {\hat{s}}_{1}\left(n\right) & \cdots & {\hat{s}}_{L_{s}-1}\left(n\right) \end{array}\right]}^{T}$; ${\hat{s}}_{l}\left(n\right)$ is $l^{\mathrm{th}}$ the coefficient of $\hat{S}\left(z\right)$ with length $L_{s}$; and $a\left(n\right)={\left[a\left(n\right)\;a\left(n-1\right)\cdots a\left(n-L_{s}+1\right)\right]}^{T}$. The audio-interference cancellation filter $\hat{S}\left(z\right)$ is an estimate of secondary-path transfer function S(z). Then, the unwanted audio component $a^{\prime }\left(n\right)$ is subtracted from the error signal e(n), expressed as
(5)$$e\left(n\right)=d\left(n\right)-u^{\prime }\left(n\right)=d\left(n\right)-\left[y^{\prime }\left(n\right)+a^{\prime }\left(n\right)\right],$$

If the audio-cancellation filter can estimate the secondary-path perfectly, such that $\hat{S}\left(z\right)=S\left(z\right)$, then ${\hat{a}}^{\prime }\left(n\right)≅a^{\prime }\left(n\right).$ Therefore, we get the residual noise signal
(6)$$e^{\prime }\left(n\right)=d\left(n\right)-y^{\prime }\left(n\right),$$
to update W(z) using the FXLMS algorithm given by
(7)$$w\left(n+1\right)=w\left(n\right)+\mu_{w}x^{\prime }\left(n\right)e^{\prime }\left(n\right),$$
where $\mu_{w}$ is the step size and $w\left(n\right)={\left[w_{0}\left(n\right)\;w_{1}\left(n\right)\cdots w_{L-1}\left(n\right)\right]}^{T}$. In Equation (7), $x^{\prime }\left(n\right)={\left[x^{\prime }\left(n\right)\;x^{\prime }\left(n-1\right)\cdots x^{\prime }\left(n-L+1\right)\right]}^{T}$ is the filtered reference signal given by
(8)$$x^{\prime }\left(n\right)={\hat{s}}^{T}\left(n\right)x\left(n\right).$$

The synthesized reference signal x(n) is obtained by
(9)$$x\left(n\right)=e^{\prime }\left(n\right)+y^{\prime }\left(n\right),$$
and $y^{\prime }\left(n\right)$ is the filtered output signal expressed as
(10)$$y^{\prime }\left(n\right)={\hat{s}}^{T}\left(n\right)y\left(n\right),$$
where $y\left(n\right)={\left[\begin{array}{cccc} y\left(n\right) & y\left(n-1\right) & \cdots & y\left(n-L_{s}+1\right) \end{array}\right]}^{T}$ is the output signal vector.

The audio-interference cancellation filter $\hat{S}\left(z\right)$ is obtained by on-line modeling using the audio signal as a reference signal, and is updated by the LMS algorithm given by
(11)$$\hat{s}\left(n+1\right)=\hat{s}\left(n\right)-\mu_{s}a\left(n\right)e^{\prime }\left(n\right),$$
where $a\left(n\right)={\left[\begin{array}{cccc} a\left(n\right) & a\left(n-1\right) & \cdots & a\left(n-L_{s}+1\right) \end{array}\right]}^{T}$, and $\mu_{s}$ is the step size. Note that the sign of $e^{\prime }\left(n\right)$ is inverted in the adaptation equation to compensate for the assumed subtractive secondary-path S(z).

To obtain a good estimate of the secondary-path S(z) during on-line modeling, the audio signal must be a persistent excitation and uncorrelated with the primary noise [26], which is not always present in real conditions. Therefore, the off-line secondary-path modeling, shown in Figure 2, was applied prior to the operation of the AFANC system. The white noise v(n) is the input for the secondary-path S(z) and the adaptive FIR filter $\hat{S}\left(z\right)$. The error signal is expressed as

(12)$$e_{o}\left(n\right)=d_{o}\left(n\right)-y_{o}\left(n\right),$$
and the filter $\hat{S}\left(z\right)$ is adapted using the LMS algorithm given by
(13)$$\hat{s}\left(n+1\right)=\hat{s}\left(n\right)-\mu_{o}v\left(n\right)e_{o}\left(n\right),$$
where $v\left(n\right)={\left[\begin{array}{cccc} v\left(n\right) & v\left(n-1\right) & \cdots & v\left(n-L_{s}+1\right) \end{array}\right]}^{T}$, and $\mu_{o}$ is the step size. Since the conventional AFANC system uses single-rate processing, the ANC filter W(z) and the estimated secondary-path filter $\hat{S}\left(z\right)$ are fullband filters, which require long filter lengths to cover the bandwidth. Therefore, computational complexity is high in the conventional AFANC system.

### 2.2. Proposed Algorithm Using Decimated-band Adaptive Filters (DAFs)

In order to reduce computational complexity, this paper proposed a multirate AFANC system using DAFs with short filter lengths. The proposed multirate AFANC system uses two fixed sampling rates, the higher one $f_{s}$ and the lower one $f_{anc}$ [30]. The sampling rate was changed by employing two decimators and one interpolator, as illustrated in Figure 3, with the same decimation and interpolation factors D=I. The sampling rate relation is given by
(14)$$f_{anc}=\frac{f_{s}}{D}.$$

The decimators and interpolator are coupled with lowpass filters H(z) with cut-off frequency $f_{c}\leq f_{anc}/2$. Therefore, the decimated signals had a low frequency range of 0 to $f_{anc}/2$, while the fullband signals had a frequency range of 0 to $f_{s}/2$. The high sampling rate $f_{s}$ is required for audio signal a(n) containing the high frequency component. The lower sampling frequency $f_{anc}$ is used for the $W_{D}\left(z\right)$ and ${\hat{S}}_{D}\left(z\right)$ filters’ input and output decimated-band signal processing. The factor D was chosen so that the upper limit of the decimated-band can cover the target unwanted narrowband noise frequency. The notation k was used for the time index at low sampling rate $f_{anc}$, and notation n was used for the time index at high sampling rate $f_{s}$.

The proposed AFANC system mixes the fullband audio signal a(n) and anti-noise signal y(n) in high sampling rate expressed as
(15)u(n)=y(n)+a(n),
where u(n) will drive the secondary loudspeaker to generate the anti-noise and audio sounds. After the driving signal u(n) passes though the secondary-path S(z), and combines with the noise signal d(n), the fullband error signal e(n) is obtained. Both fullband signals e(n) and a(n) were decimated into decimated-band signals $e_{D}\left(k\right)$ and $a_{D}\left(k\right)$, respectively. The details of the decimation processes are given in Appendix A.

The decimated-band error signal $e_{D}\left(k\right)$ also contains the residual noise and audio component. Therefore, the decimated-band audio-cancelation algorithm was also proposed to remove the decimated-band audio-interference component. The algorithm employs a decimated-band audio-interference cancelation filter ${\hat{S}}_{D}\left(z\right)$ with the decimated-band audio signal $a_{D}\left(k\right)$ as reference input to generate output signal as
(16)$${\hat{a}}_{D}^{\prime }\left(k\right)={\hat{s}}_{D}{\;}^{T}\left(k\right)a_{D}\left(k\right),$$
where ${\hat{s}}_{D}\left(k\right)={\left[\begin{array}{cccc} {\hat{s}}_{D0}\left(k\right) & {\hat{s}}_{D1}\left(k\right) & \cdots & {\hat{s}}_{DL_{sD}-1}\left(k\right) \end{array}\right]}^{T}$; ${\hat{s}}_{Dl}\left(k\right)$ is $l^{\mathrm{th}}$ the coefficient of ${\hat{S}}_{D}\left(z\right)$ with filter length $L_{sD}$; and $a_{D}\left(k\right)={\left[\begin{array}{cccc} a_{D}\left(k\right) & a_{D}\left(k-1\right) & \cdots & a_{D}\left(k-L_{sD}+1\right) \end{array}\right]}^{T}$. 

In the conventional system, there is no decimation process. However, the audio signal a(n) of the proposed system passes through additional decimation process, which means ${\hat{S}}_{D}\left(z\right)$ will not obtain the estimate of S(z) in fullband. On the other hand, the audio signal a(n) also passes the decimation process to obtain decimated-band audio signal $a_{D}\left(k\right)$. Since both reference and error signals of ${\hat{S}}_{D}\left(z\right)$ are decimated-band signals, covering the frequency portion from 0 to $f_{anc}/2$, thus ${\hat{S}}_{D}\left(z\right)$ is a decimated-band estimate of S(z), also with a frequency portion from 0 to $f_{anc}/2$. Therefore, the decimated-band residual noise signal $e_{D}^{\prime }\left(k\right)$ is obtained by
(17)$$e_{D}^{\prime }\left(k\right)=e_{D}\left(k\right)+{\hat{a}}_{D}^{\prime }\left(k\right).$$

As in the conventional system, the ${\hat{s}}_{D}\left(k\right)$ can be updated during on-line operation using LMS algorithm expressed as
(18)$${\hat{s}}_{D}\left(k+1\right)={\hat{s}}_{D}\left(k\right)-\mu_{sD}a_{D}\left(k\right)e_{D}^{\prime }\left(k\right),$$
where $a_{D}\left(k\right)={\left[\begin{array}{cccc} a_{D}\left(k\right) & a\left(k-1\right) & \cdots & a\left(k-L_{sD}+1\right) \end{array}\right]}^{T}$ and $\mu_{sD}$ is the step size.

The decimated-band residual noise signal $e_{D}^{\prime }\left(k\right)$ was used to update the decimated-band ANC filter $W_{D}\left(z\right)$ using the decimated-band FXLMS algorithm expressed as
(19)$$w_{D}\left(k+1\right)=w_{D}\left(k\right)+\mu_{wD}x_{D}^{\prime }\left(k\right)e_{D}^{\prime }\left(k\right),$$
where $w_{D}\left(k\right)={\left[\begin{array}{cccc} w_{D0}\left(k\right) & w_{D1}\left(k\right) & \cdots & w_{DL_{D}-1}\left(k\right) \end{array}\right]}^{T}$, $w_{Dl}\left(k\right)$ is $l^{\mathrm{th}}$ the coefficient of $W_{D}\left(z\right)$ with filter length $L_{D}$, $x_{D}^{\prime }\left(k\right)={\left[\begin{array}{cccc} x_{D}^{\prime }\left(k\right) & x_{D}^{\prime }\left(k-1\right) & \cdots & x_{D}^{\prime }\left(k-L_{D}+1\right) \end{array}\right]}^{T}$ with $x_{D}^{\prime }\left(k\right)$ is the filtered decimated-band reference signal and $\mu_{wD}$ is the step size.

To obtain the filtered decimated-band reference signal $x_{D}^{\prime }\left(k\right)$, the transfer function of multirate secondary-path $S_{M}\left(z\right)$ between $y_{D}\left(k\right)$ to $e_{D}^{\prime }\left(k\right)$ is required, which includes the interpolator, lowpass filter H(z), secondary-path S(z), lowpass filter H(z), and decimator. Defining the transfer function of the interpolator, cascade of the two lowpass filters, and the decimator as multirate transfer function M(z), we get $S_{M}\left(z\right)=M\left(z\right)S\left(z\right)$. Since $x_{D}^{\prime }\left(k\right)$ is in a decimated-band, only the decimated-band estimate of $S_{M}\left(z\right)$ is required, in which the decimated-band estimate of S(z) is the copy of the audio-interference cancelation filter ${\hat{S}}_{D}\left(z\right)$. The estimate of multirate transfer function $\hat{M}\left(z\right)$ can be estimated by the proposed off-line multirate transfer function estimation algorithm presented in Section 2.2.1. Therefore, the decimated-band estimate of $S_{M}\left(z\right)$ is the cascade of $\hat{M}\left(z\right){\hat{S}}_{D}\left(z\right)$. Therefore, the filtered decimated-band reference signal $x_{D}^{\prime }\left(k\right)$ can be obtained by filtering synthesized reference signal $x_{D}\left(k\right)$ with $\hat{M}\left(z\right)$ and ${\hat{S}}_{D}\left(z\right)$ expressed as
(20)$$x_{m}\left(k\right)={\hat{m}}^{T}x_{D}\left(k\right),$$
(21)$$x_{D}^{\prime }\left(k\right)={\hat{s}}_{D}{\;}^{T}\left(k\right)x_{m}\left(k\right),$$
where $x_{D}\left(k\right)={\left[\begin{array}{cccc} x_{D}\left(k\right) & x_{D}\left(k-1\right) & \cdots & x_{D}\left(k-L_{m}+1\right) \end{array}\right]}^{T}$, $\hat{m}={\left[\begin{array}{cccc} {\hat{m}}_{0} & {\hat{m}}_{1} & \cdots & {\hat{m}}_{L_{m}-1} \end{array}\right]}^{T}$; ${\hat{m}}_{l}$ is $l^{\mathrm{th}}$ the coefficient of $\hat{M}\left(z\right)$ with filter length $L_{m}$; and $x_{m}\left(k\right)={\left[\begin{array}{cccc} x_{m}\left(k\right) & x_{m}\left(k-1\right) & \cdots & x_{m}\left(k-L_{sD}+1\right) \end{array}\right]}^{T}$.

The decimated-band reference signal $x_{D}\left(k\right)$ is synthesized using the decimated-band residual noise signal $e_{D}^{\prime }\left(k\right)$ expressed as
(22)$$x_{D}\left(k\right)=e_{D}^{\prime }\left(k\right)+y_{D}^{\prime }\left(k\right),$$
where $y_{D}^{\prime }\left(k\right)$ is the filtered decimated-band anti-noise signal. Since the decimated-band anti-noise signal $y_{D}\left(k\right)$ passes through the transfer function between $y_{D}\left(k\right)$ to $e_{D}^{\prime }\left(k\right)$, thus $y_{D}\left(k\right)$ is also filtered by $\hat{M}\left(z\right)$ and ${\hat{S}}_{D}\left(z\right)$, expressed as
(23)$$y_{m}\left(k\right)={\hat{m}}^{T}y_{D}\left(k\right),$$
(24)$$y_{D}^{\prime }\left(k\right)={\hat{s}}_{D}{\;}^{T}\left(k\right)y_{m}\left(k\right),$$
where $y_{D}\left(k\right)={\left[\begin{array}{cccc} y_{D}\left(k\right) & y_{D}\left(k-1\right) & \cdots & y_{D}\left(k-L_{m}+1\right) \end{array}\right]}^{T}$ and $y_{m}\left(k\right)={\left[\begin{array}{cccc} y_{m}\left(k\right) & y_{m}\left(k-1\right) & \cdots & y_{m}\left(k-L_{sD}+1\right) \end{array}\right]}^{T}$. Using the synthesized decimated-band reference signal $x_{D}\left(k\right)$, the decimated-band adaptive filters $W_{D}\left(z\right)$ generate the decimated-band anti-noise signal $y_{D}\left(k\right)$ expressed as
(25)$$y_{D}\left(k\right)=w_{D}{\;}^{T}\left(k\right)x_{D}\left(k\right),$$
where $x_{D}\left(k\right)={\left[\begin{array}{cccc} x_{D}\left(k\right) & x_{D}\left(k-1\right) & \cdots & x_{D}\left(k-L_{D}+1\right) \end{array}\right]}^{T}$. Then, the anti-noise signal $y_{D}\left(k\right)$ is interpolated to fullband output signal y(n) to drive the secondary loudspeaker. The interpolation process is given in Appendix A.

To obtain a good estimate of the secondary-path S(z) during on-line modeling, the audio signal must be a persistent excitation and uncorrelated with the primary noise [26], which is not always present in real conditions. When a suitable audio signal is not available, the estimated decimated-band secondary-path model can be obtained by off-line operation, prior to the operation of the audio-integrated ANC. Therefore, the decimated-band off-line modeling algorithm was proposed, as shown in Figure 4.

The white noise signal v(n) in fullband frequency range is used to drive the secondary-path S(z) and $d_{o}\left(n\right)$ is obtained. Both signals v(n) and $d_{o}\left(n\right)$ are decimated into decimated-band signals $v_{D}\left(k\right)$ and $d_{Do}\left(k\right)$, respectively, where details are given in Appendix A. The decimated-band signal $v_{D}\left(k\right)$ was used as the reference input signal for the decimated-band adaptive FIR filter ${\hat{S}}_{D}\left(z\right)$ to generate the output signal given by
(26)$$y_{Do}\left(k\right)={\hat{s}}_{D}{\;}^{T}\left(k\right)v_{D}\left(k\right),$$
where ${\hat{s}}_{D}\left(k\right)={\left[\begin{array}{cccc} {\hat{s}}_{D0}\left(k\right) & {\hat{s}}_{D1}\left(k\right) & \cdots & {\hat{s}}_{DL_{sD}-1}\left(k\right) \end{array}\right]}^{T}$; ${\hat{s}}_{Dl}\left(k\right)$ is $l^{\mathrm{th}}$ the coefficient of ${\hat{S}}_{D}\left(z\right)$ with filter length $L_{sD}$; and $v_{D}\left(k\right)={\left[\begin{array}{cccc} v_{D}\left(k\right) & v_{D}\left(k-1\right) & \cdots & v_{D}\left(k-L_{sD}+1\right) \end{array}\right]}^{T}$. Therefore, the decimated-band error signal is
(27)$$e_{Do}\left(k\right)=d_{Do}\left(k\right)-y_{Do}\left(k\right).$$

The decimated-band error signal $e_{Do}\left(k\right)$ was used to update the coefficients of ${\hat{S}}_{D}\left(z\right)$ using the decimated-band LMS algorithm given by
(28)$${\hat{s}}_{D}\left(k+1\right)={\hat{s}}_{D}\left(k\right)-\mu_{Do}v_{D}\left(k\right)e_{Do}\left(k\right).$$
where $\mu_{Do}$ is the step size.

#### 2.2.1. Off-Line Multirate Transfer Function Modeling Algorithm

The decimated-band estimate of multirate transfer function $\hat{M}\left(z\right)$ is required for an ANC system due to the multirate processing. Since the multirate transfer function M(z) is fixed, the decimated-band estimate of one lowpass filter H(z) can be calculated directly. However, it is much more complicated to directly calculate the decimated-band estimate of the overall multirate transfer function M(z). Therefore, it can be estimated by off-line operation prior to the operation of the proposed AFANC system using the proposed off-line multirate transfer function modeling algorithm shown in Figure 5. The filter $\hat{M}\left(z\right)$ is a decimated-band estimate of the interpolator, lowpass filter H(z), decimator, and lowpass filter H(z). 

The white noise signal $v_{Dm}\left(k\right)$ is in the low sampling rate $f_{anc}$, and is used as the reference signal for the adaptive FIR filter $\hat{M}\left(z\right)$ for generating the output signal expressed as
(29)$$y_{mo}\left(k\right)={\hat{m}}^{T}\left(k\right)v_{Dm}\left(k\right),$$
where $\hat{m}\left(k\right)={\left[\begin{array}{cccc} {\hat{m}}_{0}\left(k\right) & {\hat{m}}_{1}\left(k\right) & \cdots & {\hat{m}}_{L_{m}-1}\left(k\right) \end{array}\right]}^{T}$; ${\hat{m}}_{l}\left(k\right)$ is the $l^{\mathrm{th}}$ coefficient of $\hat{M}\left(z\right)$ with filter length $L_{m}$; and $v_{Dm}\left(k\right)={\left[\begin{array}{cccc} v_{Dm}\left(k\right) & v_{Dm}\left(k-1\right) & \cdots & v_{Dm}\left(k-L_{m}+1\right) \end{array}\right]}^{T}$. The white noise signal $v_{Dm}\left(k\right)$ passes through interpolator, two lowpass filters, and decimators, and $d_{m}\left(k\right)$ is obtained. The equations to obtain $d_{m}\left(k\right)$ are given in Appendix A.

The decimated-band error signal $e_{mo}\left(k\right)$ is given by
(30)$$e_{mo}\left(k\right)=d_{mo}\left(k\right)-y_{mo}\left(k\right).$$
and used to update the coefficients of $\hat{M}\left(z\right)$,
(31)$$\hat{m}\left(k+1\right)=\hat{m}\left(k\right)-\mu_{m}v_{Dm}\left(k\right)e_{mo}\left(k\right).$$
where $\mu_{m}$ is the step size. After convergence is achieved, the coefficients of $\hat{M}\left(z\right)$ are fixed and were used for the on-line proposed multirate AFANC system operation.

### 2.3. Performance Analysis of Conventional and Proposed AFANC Systems

The use of bandpass filters will add additional delay on the signal path, which limits the controllable bandwidth in the proposed system. However, the delay would not bring much disadvantages on reducing narrowband noises [15]. The main motivation of using DAFs in the proposed system is to reduce the computational complexity with moderate convergence speed in reducing narrowband noises. The computational complexity is influenced by the number of filters and their lengths. The convergence speed is influenced by the step size. 

#### 2.3.1. Computational Complexity

In general, a filter length of fullband filter can be reduced by decimation factor D in decimated-band filter [31]. Assuming the filter lengths of W(z) and $\hat{S}\left(z\right)$ are L and $L_{s}$, respectively, for the conventional system; thus, the filter lengths of $W_{D}\left(z\right)$ and ${\hat{S}}_{D}\left(z\right)$ are
(32)$$L_{D}=L/D,\;\mathrm{and}$$
(33)$$L_{sD}=L_{s}/D,$$
respectively. Furthermore, let the filter length for H(z) be $L_{h}$, thus, the filter length of $\hat{M}\left(z\right)$ is
(34)$$L_{m}=2L_{h}/D,$$
in the proposed system. 

The conventional AFANC system uses one W(z) and three $\hat{S}\left(z\right)$, and the computation for these filters output and adaptations was $2\left(L+2L_{s}\right)+2$ multiplications and $2\left(L+2L_{s}\right)-4$. The proposed system used one $W_{D}\left(z\right)$, three ${\hat{S}}_{D}\left(z\right)$, three H(z), and two $\hat{M}\left(z\right)$, and the computation for these filters’ output and adaptations were $2\left(L_{D}+2L_{sD}+3.5L_{h}/D\right)+2$ multiplications and $2\left(L_{D}+2L_{sD}+3.5L_{h}/D\right)-9$ additions. Table 1 summarizes the computational complexity in the conventional and proposed systems. 

From Table 1, we can see that the required computation depends on the L, $L_{s}$, $L_{h}$, and D. In practice, the choices of L, $L_{s}$, and D are influenced by noise and secondary-path. However, $L_{h}$ is designed by the user. The longer the $L_{h}$, the better the lowpass filter response in having flat passband, sharp cut-off frequency, and large stopband attenuation. Taking L=800, $L_{s}=256$, $L_{h}=80$, and D=10 as an example, the conventional system requires 2626 multiplications and 2620 additions, and the proposed system requires 538 multiplications and 527 additions, in which significant computational reduction can be achieved. However if $L_{h}=694$, the proposed system requires 2626 multiplications and 2617 additions, which is the upper bound in achieving the computational reduction in this case. If $L_{h}$ is higher than 694, the proposed system will have more computations. In practice, $L_{h}$ can be much less than 694.

Considering the conventional system uses a very high sampling rate such as 48 kHz, as recommended as the professional digital audio sampling rate [32], thus, the computations have to be completed in a very short period. If the computation time is not enough, the proposed system may use a very high speed digital signal processor (DSP) for implementation, or reduce the filter lengths which may degrade the performance. 

#### 2.3.2. Convergence Analysis

Decreasing the sampling rate will affect the convergence speed because a higher sampling rate system will have more adaptation iterations in the same time duration compared to a lower sampling rate system. Convergence speed is also governed by the step size, where a larger step size can usually have a faster convergence speed. The following convergence analysis derived the step size bounds for the conventional and proposed systems.

For the purpose of analysis, the secondary-path estimation was assumed to be perfect, $\hat{S}\left(z\right)=S\left(z\right)$. Assuming that audio-interference is perfectly cancelled, the step sizes $\mu_{w}$ and $\mu_{wD}$ for the W(z) and $W_{D}\left(z\right)$ adaptations can be approximated by the step size bounds of the adaptive feedback ANC system given by [33]
(35)$$\mu_{w}<\frac{2}{\left(2∆+1\right)LP_{x^{\prime }}},\;\mathrm{and}$$
(36)$$\mu_{wD}<\frac{2D}{\left(2{∆}_{D}+1\right)L_{D}P_{x_{D}{\;}^{\prime }}},\;$$
where ∆ and ${∆}_{D}$ are the delays of the secondary-path S(z) and the multirate secondary-path $S_{M}\left(z\right)$ in the conventional and proposed system, respectively. The delay estimation is given in Appendix B. The $P_{x^{\prime }}$ and $P_{x_{D}{\;}^{\prime }}$ are the power of the filtered reference signals $x^{\prime }\left(n\right)$ and $x_{D}^{\prime }\left(k\right)$, respectively.

Let
(37)$${∆}_{g}=\frac{{∆}_{D}}{∆},\;\mathrm{and}$$
(38)$$P_{g}=\frac{P_{x^{\prime }}}{P_{x_{D}{\;}^{\prime }}},$$
where ${∆}_{g}$ is delay gain; $P_{g}$ is power gain; then the step size bound $\mu_{wD}$ can be expressed as
(39)$$\mu_{wD}≅\frac{2D^{2}P_{g}}{\left(2∆{∆}_{g}+1\right)LP_{x^{\prime }}}.$$
where $\mu_{wD}$ relates to $\mu_{w}$ in terms of $P_{g}$ and ${∆}_{D}$. 

In theory, the step size determines the convergence speed of the adaptation algorithms. However, a direct step size comparison is not valid as a comparison of convergence speed due to the different sampling rates between conventional and proposed systems. Different sampling rates will give different convergence speeds in thee time unit with the same step size. Supposing that the adaptive filter W(z) achieves convergence in i-th iterations on sampling rate $f_{s}$, it means that the convergence time is $i/f_{s}$. On the other hand, if the adaptive filter $W_{D}\left(z\right)$ also converges in the i-th iterations on sampling rate $f_{anc}$, it means that the convergence time is $Di/f_{s}$. Therefore, to achieve the same convergence speed in the time unit, the step size of $W_{D}\left(z\right)$ has to achieve
(40)$$\mu_{wD}\geq D^{2}\mu_{w}.$$

In the conventional and proposed off-line secondary-path modeling, the step size bound can be derived from (35) with delay ∆=0 [34], given by
(41)$$\mu_{o}<\frac{2}{LP_{v}},\;\mathrm{and}$$
(42)$$\mu_{Do}<\frac{2}{LP_{vD}},$$
where $P_{v}$ and $P_{vD}$ are the power of the reference signals v(n) and $v_{D}\left(k\right)$, respectively. For on-line secondary-path modeling, the step sizes are smaller, $\mu_{s}\ll \mu_{o}$ and $\mu_{sD}\ll \mu_{Do}$ due to misalignment caused by disturbances from primary noises [14]. 

## 3. Results and Discussion

Simulations and real-time experiments were conducted to evaluate the performance of the proposed system.

### 3.1. Experiment Setup

The proposed AFANC system was implemented for the active headrest system, as shown in Figure 6. The TMS320C6713DSK from Texas Instrument (TI) and the analog-to-digital and digital-to-analog converter interface card from the Heg Company were utilized as the core of the proposed system. Two Tang band (T1-1828S) loudspeakers with dimensions of 13.5 × 5.5 × 2.2 cm were used as the secondary loudspeakers and two Shure MX183 microphones were applied as the error microphones. These two error microphones and secondary loudspeakers were combined as one in the analog domain as proposed in [35]. An ALESIS Mictube Duo Stereo and SMSL SA-98E were used as the microphone’s preamplifier and loudspeaker power amplifier. 

The primary noises contained 16 narrowband noises from 300 to 600 Hz equally separated by 20 Hz, combined with white noise with a narrowband and white noise ratio of 0 dB. The noise was played by a primary loudspeaker located 1 m behind the headrest. The active headrest system was tested to cancel these noises and deliver music. The recommended sampling rate for audio was 48 kHz [32]. Therefore, the sampling rate for the conventional system was $f_{s}=48$ kHz and for the proposed system, they were $f_{s}=48$ kHz and $f_{anc}=2$ kHz. Therefore, the decimation factor D is 24.

### 3.2. Computer Simulations

Computer simulations with MATLAB software were conducted to compare the performance of the conventional and proposed systems. The frequency response of the secondary path transfer function from the active headrest system was measured by a Keysight 35670A dynamic signal analyzer. Then, the frequency response was curve-fitted by the 255th order FIR filter as the S(z) model for the simulations. To compare the performances of using different filter lengths, the proposed system was simulated using two different lowpass filters $L_{h}$ with magnitude responses as shown in Figure 7. The blue line response was $L_{h}=192$ and the cutoff frequency was 694 Hz, and the green line response was $L_{h}=48$ and cutoff frequency was 711 Hz. The cutoff frequencies were less than 1 kHz to achieve the decimated-band signals with a frequency range of 0 to 1 kHz.

#### 3.2.1. Off-Line Secondary-Path Modeling Results

Conventional and proposed off-line secondary-path modeling were performed prior to on-line AFANC operations to obtain $\hat{S}\left(z\right)$ and ${\hat{S}}_{D}\left(z\right)$. The filter length of $\hat{S}\left(z\right)$ filter $L_{s}$ was 256, following the filter length of S(z). Since D=24, the corresponding filter length $L_{sD}$ of the estimated decimated-band secondary-path filter ${\hat{S}}_{D}\left(z\right)$ was 26, according to (36). The proposed modeling was run twice using a different lowpass filter in each modeling. The white noise signal power was 1. Therefore, based on (41) and (42), the step sizes for the conventional and proposed off-line modeling were $\mu_{o}=0.0039$ and $\mu_{Do}=0.0909$.

Figure 8 shows the off-line secondary-path modeling results, where the black and dashed-red lines show the magnitude and phase responses of S(z), and $\hat{S}\left(z\right)$, respectively. The blue line shows ${\hat{S}}_{D}\left(z\right)$ with $L_{h}=48$, and the green line with $L_{h}=192$. For the conventional system, the fullband frequency response is required. Since the filter length of $\hat{S}\left(z\right)$ was the same as S(z), the estimated secondary-path filter $\hat{S}\left(z\right)$ fit the fullband response of S(z) almost perfectly. 

On the other hand, the proposed system only needs the decimated-band frequency response of S(z). Figure 8 shows the frequency response of ${\hat{S}}_{D}\left(z\right)$ to estimate S(z) in a frequency range of 0–1 kHz. The magnitude responses of the lowpass filter with $L_{h}=48$ was different than with $L_{h}=192$. Therefore, ${\hat{S}}_{D}\left(z\right)$ with $L_{h}=48$ had a different response than with $L_{h}=192$. We can see that ${\hat{S}}_{D}\left(z\right)$ had a poor estimate of S(z) at aa frequency around 0–200 Hz and 0.8–1 kHz, influenced by the steep curving response of S(z) in low frequency and lowpass filter cutoff frequency below 1 kHz. However, ${\hat{S}}_{D}\left(z\right)$ still had good estimates at a frequency around 200–800 Hz. This result shows that ${\hat{S}}_{D}\left(z\right)$ with a shorter filter length can effectively estimate the decimated-band transfer function of S(z).

#### 3.2.2. Off-Line Multirate Transfer Function Modeling Results

The multirate transfer function filter $\hat{M}\left(z\right)$ is required to realize the proposed system. Since D=24, according to (34), the filter length of $\hat{M}\left(z\right)$ was $L_{m}=4$ for $L_{h}=48$, and $L_{m}=16$ for $L_{h}=192$.

Figure 9 shows the modeling results, where black lines in Figure 9a,b show the magnitude responses of multirate transfer function with $L_{h}=48$ and $L_{h}=192$, respectively. The blue line in Figure 9a and green line in Figure 9b show the magnitude responses of $\hat{M}\left(z\right)$ with $L_{m}=4$ and $L_{m}=16$, respectively. Obviously, the magnitude response of $\hat{M}\left(z\right)$ fit the corresponding magnitude response of the multirate transfer function in a frequency range of 0–1 kHz. These results show the effectiveness of the proposed off-line multirate transfer function modeling algorithm.

#### 3.2.3. Noise Reduction Results

The fullband secondary-path filter $\hat{S}\left(z\right)$ and decimated-band secondary-path filter ${\hat{S}}_{D}\left(z\right)$ obtained from the off-line modeling were used as initials in on-line AFANC operations. The fixed coefficient multirate-path filter $\hat{M}\left(z\right)$ obtained from the off-line multirate transfer function was used in the proposed AFANC system.

The first simulation focused on the noise reduction performance of the conventional and proposed AFANC systems without the existence of an audio signal. For comparison purposes, two conditions were tested for the conventional system. First with a shorter filter length L=1500, and then with a longer filter length L=2400. The proposed system used $L_{D}=100$ for $W_{D}\left(z\right)$, which satisfies (32) for a longer filter length L=2400. Two conditions for the proposed system were tested. First with a lowpass filter length $L_{h}=48$, and the second one with $L_{h}=192$.

The step sizes were calculated according to (35) and (36). The delay of the secondary-path in conventional system was 153.7. Furthermore, the delays of the multirate secondary-paths in the proposed system were 200.7 and 344.7 for $L_{h}=48$ and $L_{h}=192$, respectively. Thus, the delay gain ${∆}_{g}$ was 1.3 and 2.4 with $L_{h}=48$ and $L_{h}=192$, respectively. Therefore, we can see that using a longer lowpass filter length has more delay gain in the multirate secondary-path. 

The filtered reference signal power $P_{x^{\prime }}$ and $P_{x_{D}{\;}^{\prime }}$ were estimated by obtaining $x^{\prime }\left(n\right)$ and $x_{D}{\;}^{\prime }\left(n\right)$ prior to on-line ANC operations. Then, the mean values of all ${x^{\prime }}^{2}\left(n\right)$ and $x_{D}{{\;}^{\prime }}^{2}\left(n\right)$ samples were used as estimated $P_{x^{\prime }}$ and $P_{x_{D}{\;}^{\prime }}$, respectively. The estimated $P_{x^{\prime }}$ was 11.4, and the estimated $P_{x_{D}{\;}^{\prime }}$ were 7.5 and 9.9, for $L_{h}=48$ and $L_{h}=192$, respectively. The power with $L_{h}=192$ was higher due to its higher cutoff frequency. The power gain $P_{g}$ were 1.5 and 1.2, for $L_{h}=48$ and $L_{h}=192$, respectively. Therefore, we can see that the conventional system obtained larger filtered reference signal power compared to the proposed system.

After computing delays and filtered reference signal powers, the step size bounds were obtained, and the used step sizes were half of the bounds. The step sizes for conventional systems were $1.89\cdot 10^{-7}$ and $1.18\cdot 10^{-7}$ for L=1500 and L=2400, respectively. The step sizes for the proposed systems were $7.9\cdot 10^{-5}$ and $3.5\cdot 10^{-5}$ for $L_{D}=100$, $L_{h}=48$ and $L_{D}=100$, $L_{h}=192$, respectively.

Figure 10 shows the learning curves using the exponential window technique [14] with a smoothing parameter of 5000. Both the conventional and proposed systems were simulated for 4 s. The red and brown lines in Figure 10 show the learning curves from conventional systems with L=1500 and L=2400, respectively, and the blue and green lines show the learning curves from the proposed systems with $L_{D}=100$, $L_{h}=48$ and $L_{D}=100$, $L_{h}=192$, respectively. As can be seen from Figure 10, the conventional system with a longer filter length of L=2400 and the proposed system could achieve a mean square error (MSE) level of around 0 dB at four seconds, which is the white noise MSE level, however, the conventional system with a shorter filter length L=1500 could not achieve the same MSE level because the length was not long enough. The proposed system with $L_{D}=100$, $L_{h}=48$ had a similar learning curve as the conventional system with L=2400, which means both systems had similar convergence speed. The proposed system with $L_{D}=100$, $L_{h}=192$ almost had the same convergence speed as the conventional system with L=2400.

Comparing the step sizes used in this simulation, we could see that $\mu_{wD}\approx 669.6\mu_{w}$ in the conventional system with $L_{D}=100$, $L_{h}=48$ and the proposed system with L=2400. In this condition, although the proposed system gained a delay by ${∆}_{g}=1.3$, the conventional system had a longer filter length and more power with $P_{g}\approx 1.52$. Therefore, the same convergence condition in Equation (40) was achieved, as shown in the simulation result. However, by using longer lowpass filter length $L_{h}=192$, the step size comparison was $\mu_{wD}\approx 297.2\mu_{w}$ in the conventional system and proposed system with L=2400.

The computations required by the conventional system in each iteration were 4026 multiplications and 4020 additions with L=1500, and 5826 multiplications and 5820 additions with L=2400. The computations required for the proposed system were 458 multiplications and 447 additions with $L_{h}=48$, and 914 multiplications and 903 additions with $L_{h}=192$. The conventional system could only achieve good noise reduction when using L=2400. However, only by $L_{D}=100$ and $L_{h}=48$, the proposed system can achieve a similar performance with 12 times computational reduction.

### 3.3. Real-Time Experiments

Real-time experiments were conducted to compare the performance of the conventional and proposed systems in reducing noises and delivering audio sound in real conditions. The primary noises, lowpass filter H(z), and multirate filter M(z) were the same as in the simulation. The filter lengths of W(z) and $\hat{S}\left(z\right)$ in the conventional system were L=400 and $L_{s}=150$, respectively, and the filter lengths of $W_{D}\left(z\right)$ and ${\hat{S}}_{D}\left(z\right)$ in the proposed system were $L_{D}=200$ and $L_{sD}=110$, respectively.

In the first experiment, sinusoid signal at frequency 700 Hz was used as the audio signal to see whether the audio sound was not canceled by the AFANC system. The primary noise was the same as defined in Section 3.1. Figure 11 shows the spectrums of the error signals measured by the error microphones, where the black line shows the spectrum when ANC is off, and the red and blue lines are from the conventional and proposed systems, respectively, when the ANC is on. Obviously, the proposed system achieved better narrowband noise reduction compared to the conventional system. In addition, the conventional system also increased the broadband noise. The audio sound at 700 Hz was maintained in the conventional and proposed systems. 

In the second experiment, non-stationary real music was used as the audio signal, and primary noises were the same as the first experiment. Figure 12 shows the spectrogram of the recorded error microphone signal, where the horizontal and vertical axes show the time and frequency, respectively, and color shows the signal power. As can be seen, only primary noises were present from the beginning to five seconds, when the ANC was off. The red line at 300–600 Hz represents the present narrowband component of primary noises, and the yellow-red background color represents the white noise. At 5 to 15 s, the music audio was fed to the system with the ANC off. The yellow-red background color was darker, indicating that a music component was added. Then, the ANC of the conventional system was turned on at 15 s to 25 s, and from 25 s to 35 was the proposed system with ANC on. Obviously, the red lines at 300–600 Hz from 25 to 35 s were almost not seen, indicating that the proposed system achieved good narrowband noise reduction. However, the red lines at 300–600 Hz from 15 to 25 s in the conventional system were still clearly seen, which means that the narrowband noises were not reduced as well as in the proposed system.

The experiment results showed that the conventional AFANC system with longer fullband filters could not achieve good enough noise reduction, and that improvement by increasing the filter length was also impractical. However, the proposed multi-rate AFANC system achieved better noise reduction and effectively delivered music using a sampling frequency of 48 kHz by employing decimated-band adaptive filters with shorter filter lengths. 

## 4. Conclusions

This paper presented the conventional and proposed AFANC systems to achieve low frequency narrowband noise reduction and deliver audio sound in a high sampling rate. The conventional system uses single-rate processing, thus the fullband adaptive filters for ANC and audio-interference cancelation require long filter lengths, resulting in high computational complexity. Therefore, this paper proposed a multirate AFANC system utilizing two decimators to obtain the decimated-band signals used for ANC and audio-interference cancelation. Hence, the adaptive filters worked on the decimated-band with shorter filter lengths. One interpolator was employed to convert the decimated-band anti-noise signal into a fullband signal for achieving audio signal mixing with high sampling rate processing.

The computational complexity and convergence of both systems were analyzed. Since the filter lengths can be reduced for decimated-band filters, significant computational reduction can be achieved by the proposed system if the lowpass filter length is short enough. The step size bounds of both systems were analyzed to study the convergence speed. Although the additional delay introduced in the proposed system will limit the step size bound, a similar convergence speed is still possible to achieve. Computer simulations verified that long filters are required for the conventional system to achieve good enough noise reduction; however, the proposed system could achieve good noise reduction and fast convergence speed with much shorter filter lengths, and significant computational reduction was attained. Both systems were implemented for an active headrest system and were tested in real-time experiments. The proposed system worked effectively in real conditions, while the conventional system is impractical to achieve good performance.

The proposed multirate processing comes with additional delay on the signal path. However, since the adaptive feedback ANC system is typically applied to reduce narrowband noises, the delay would not have much of an impact. To reduce broadband noise with audio integration in a high sampling rate, an audio-integrated adaptive feedforward ANC system with low computational complexity should be developed in future work.

## Figures and Tables

**Figure 1 sensors-20-06693-f001:**
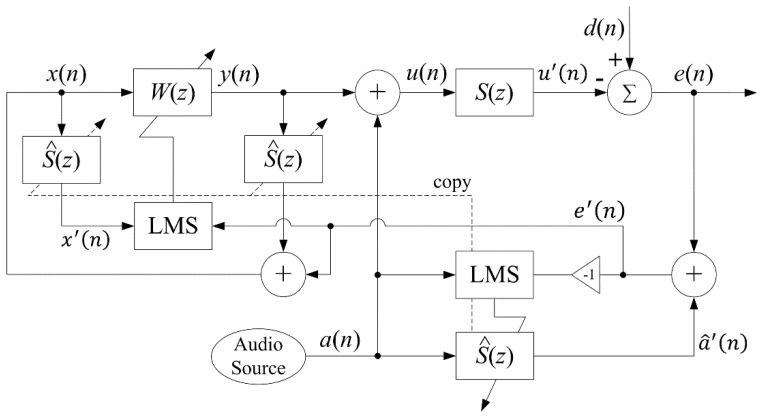
Conventional audio-integrated feedback active noise control (AFANC) system.

**Figure 2 sensors-20-06693-f002:**
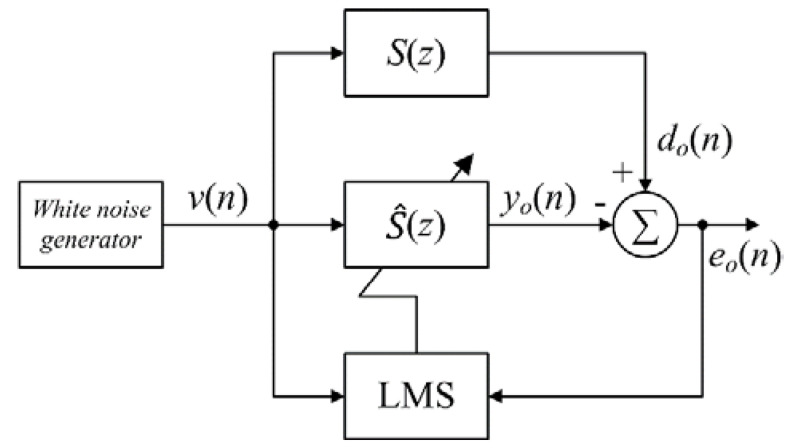
Conventional off-line modeling algorithm to estimate S(z).

**Figure 3 sensors-20-06693-f003:**
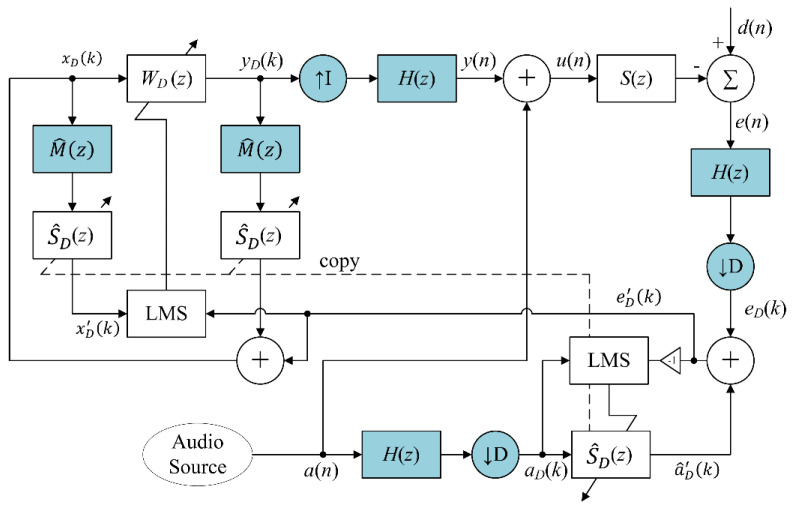
Proposed multirate audio-integrated feedback active noise control (AFANC) system using decimated-band adaptive filters (DAFs).

**Figure 4 sensors-20-06693-f004:**
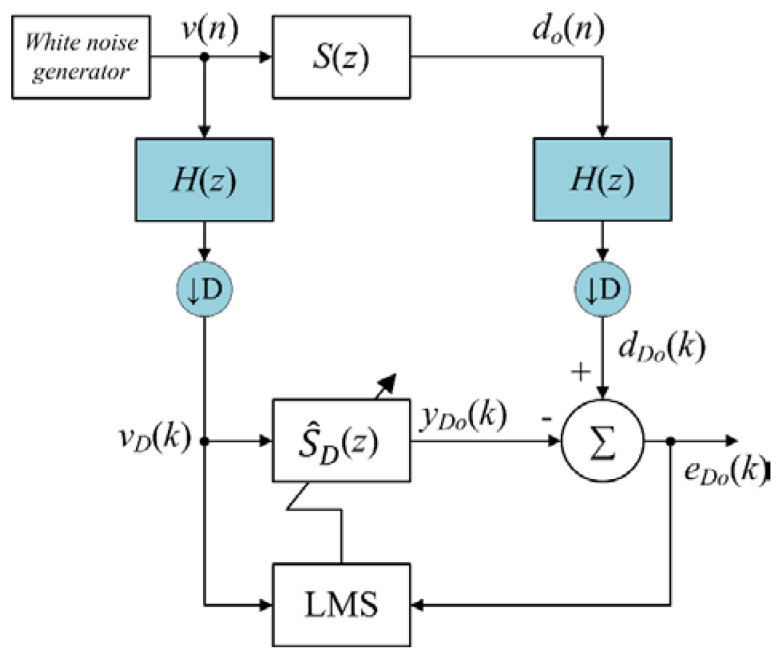
Proposed off-line decimated-band modeling algorithm to estimate the decimated-band secondary-path S(z).

**Figure 5 sensors-20-06693-f005:**
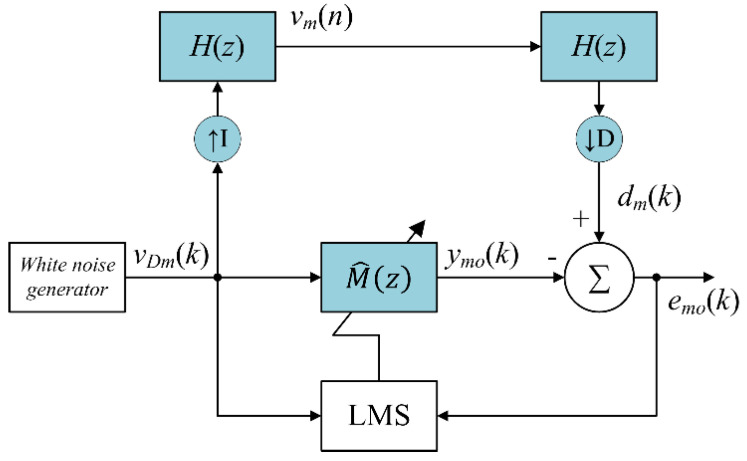
Proposed off-line multirate transfer function modeling algorithm.

**Figure 6 sensors-20-06693-f006:**
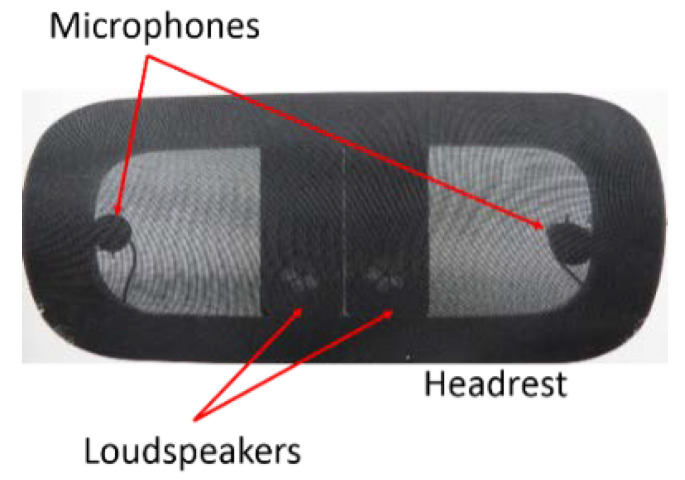
Active headrest system used for the experiments.

**Figure 7 sensors-20-06693-f007:**
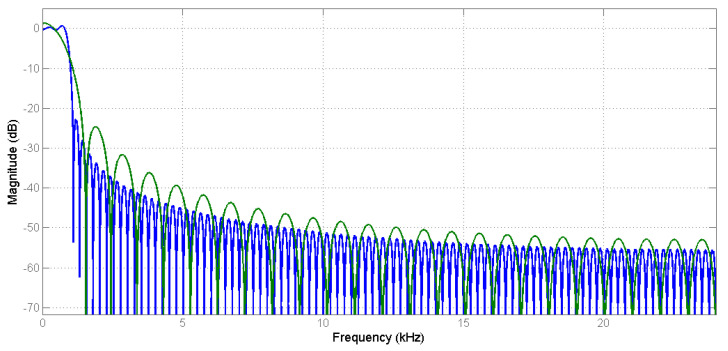
Magnitude response of lowpass filters.

**Figure 8 sensors-20-06693-f008:**
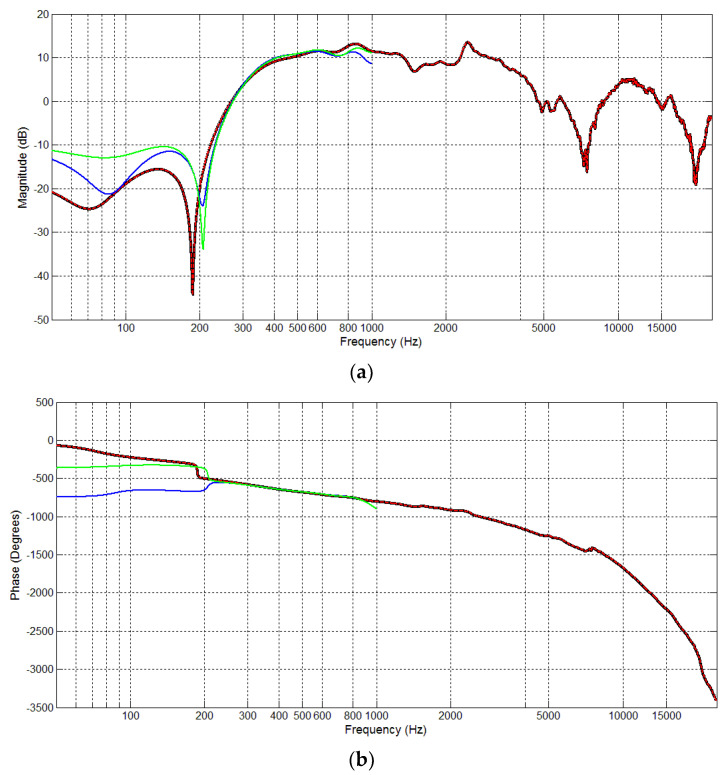
Comparison of S(z) (black), $\hat{S}\left(z\right)$ (dashed-red), ${\hat{S}}_{D}\left(z\right)$ with lowpass filter length 48 (blue), and ${\hat{S}}_{D}\left(z\right)$ with lowpass filter length 192 (green): (**a**) magnitude response and (**b**) phase response.

**Figure 9 sensors-20-06693-f009:**
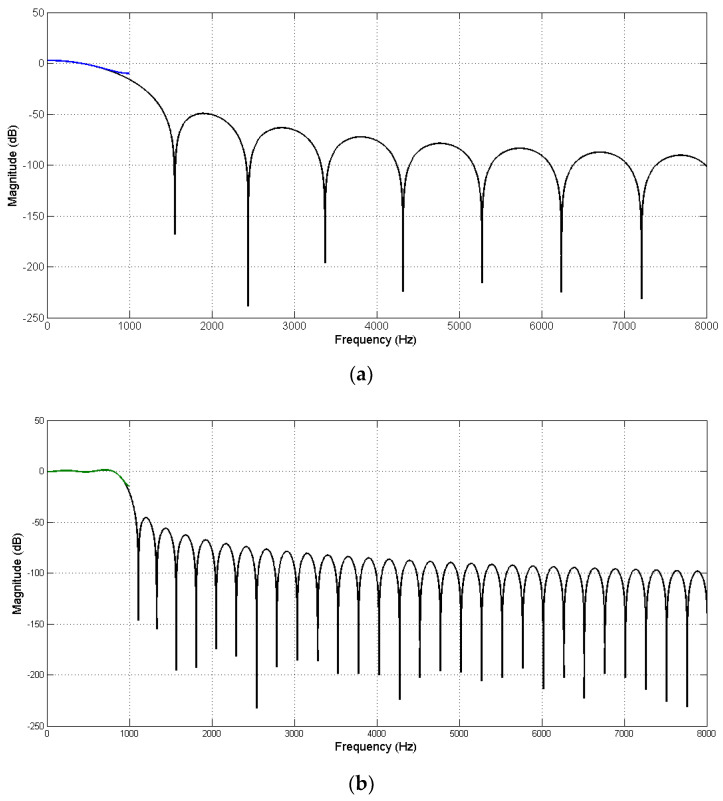
Magnitude response of the multirate transfer function (black line) compared to (**a**) $\hat{M}\left(z\right)$ (blue line) with $L_{m}=4$ and $L_{h}=48$ and (**b**) $\hat{M}\left(z\right)$ (green line) with $L_{m}=16$ and $L_{h}=192$.

**Figure 10 sensors-20-06693-f010:**
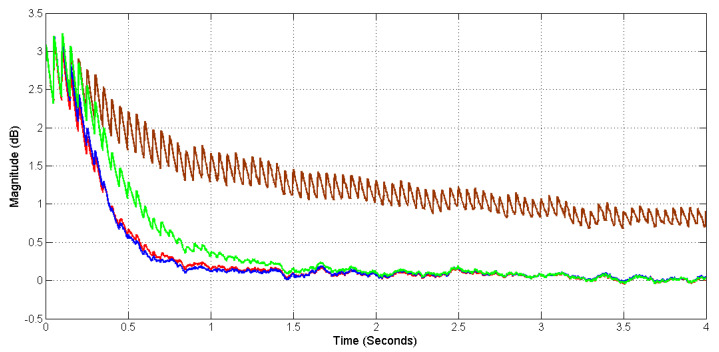
Mean square error (MSE) learning curves from conventional systems with L=1500 (brown), L=2400 (red), and proposed systems with $L_{h}=48$ (blue) and $L_{h}=192$ (green).

**Figure 11 sensors-20-06693-f011:**
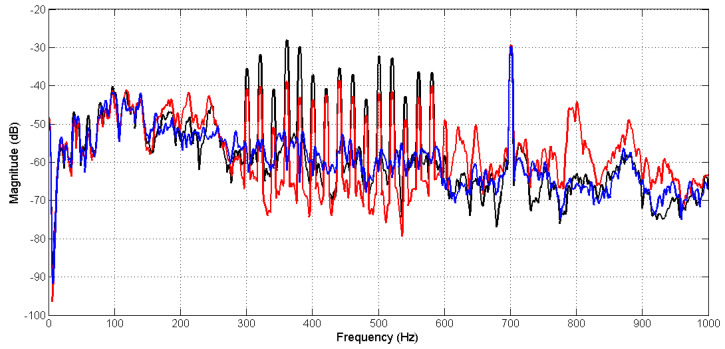
Spectrum signal measured from error microphones when the active noise control (ANC) is off (black line), and active noise control (ANC) is on (blue for the proposed and the red for the conventional methods).

**Figure 12 sensors-20-06693-f012:**
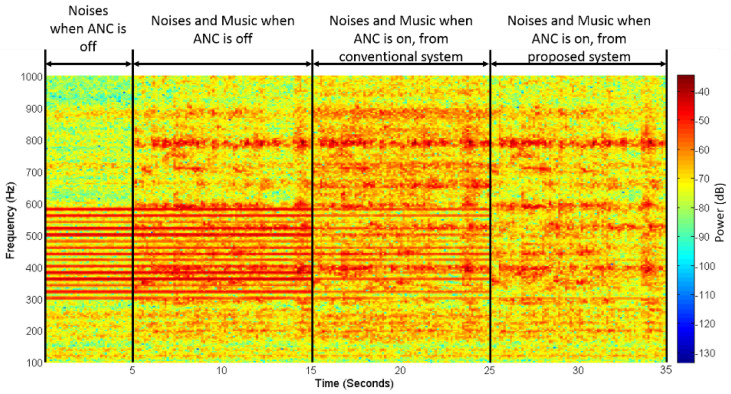
Spectrogram of error for the microphone signal.

**Table 1 sensors-20-06693-t001:** Computational complexity of the conventional and proposed systems.

AFANC System	Filter	Length	Computation	
			Multiplications	Additions
Conventional	W(z)	L	$2\left(L+2L_{s}\right)+2$	$2\left(L+2L_{s}\right)-4$
	$\hat{S}\left(z\right)$	$L_{s}$		
Proposed	$W_{D}\left(z\right)$	$L_{D}=L/D$	$2\left(L_{D}+2L_{sD}+3.5L_{h}/D\right)+2$	$2\left(L_{D}+2L_{sD}+3.5L_{h}/D\right)-9$
	${\hat{S}}_{D}\left(z\right)$	$L_{sD}=L_{s}/D$		
	H(z)	$L_{h}$		
	$\hat{M}\left(z\right)$	$L_{m}=2L_{h}/D$		

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
