# Peer review of "Multirate Audio-Integrated Feedback Active Noise Control Systems Using Decimated-Band Adaptive Filters for Reducing Narrowband Noisesâ€"

_sensors, 2020, doi:10.3390/s20226693_

Round 1
Reviewer 1 Report
This manuscript focues an audio-integrated feedback ANC system based on multi-rate signal processing for the purpose of lowering the computational burden. The motivation is reasonable but some crucial issues are not well addressed.
1) An extremely long delay is introduced into the feedback path, and this would significantly influence the performance of the whole system, such as the effective bandwidth and the waterbed effect. More discussion on this problem is necessary.
2) In the simulation, the conventional algorithm works with long enough control filter. However, its performance is much poorer than the proposed algorithm with respect to both the residual noise and the convergence speed. Though the bounds of step size have been discussed in Section 2.3.2, it is not enough to explain the poor performance of the conventional algorithm. More explanation is necessary.
3) M(z) can be calculated directly by H(z), and the estimation of M(z) with LMS algorithm seems unnecessary.
4) The manuscript in general is not well presented, with any grammatical and typo errors. A few are listed as follows.
(1) Line 27, “noises producing by rotating machines” should be “noises produced by rotating machines.”
(2) Line 239, “depend on” should be “depends on.”
(3) Line 254, “2.3.1” should be “2.3.2.”
(4) Some abbreviations appear without the corresponding explanations.
Author Response
Response to the reviewers
We would like to thank the editor’s and reviewers’ comments, which are all considered in the revised version of the manuscript to improve its quality. The modifications are highlighted using color words corresponding to the reviewers in the revised manuscript. The responses are summarized as follows:
Reviewer 1:
This manuscript focuses an audio-integrated feedback ANC system based on multi-rate signal processing for the purpose of lowering the computational burden. The motivation is reasonable but some crucial issues are not well addressed.
1) An extremely long delay is introduced into the feedback path, and this would significantly influence the performance of the whole system, such as the effective bandwidth and the waterbed effect. More discussion on this problem is necessary.
Authors’ response:
It is true that additional delay is introduced into the feedback path due to the employed lowpass filters. However, the adaptive feedback ANC system is typically used for narrowband noise reduction, which is also the aim of this paper. So, the effective bandwidth would not be the issue in this case. The authors have addressed this issue in the revised paper on lines 264-266, and 511-513. Besides, the waterbed effect occurs in negative feedback control system [15]. But the AFANC system in this paper is based on internal model control, which is equivalent to feedforward controller [14,15]. This explanation has been added into lines 50-56.
2) In the simulation, the conventional algorithm works with long enough control filter. However, its performance is much poorer than the proposed algorithm with respect to both the residual noise and the convergence speed. Though the bounds of step size have been discussed in Section 2.3.2, it is not enough to explain the poor performance of the conventional algorithm. More explanation is necessary.
Authors’ response:
In the simulation, two conditions were tested for the conventional system. First is with shorter filter length, and second is for long enough filter length. The conventional system performed poorer for shorter filter length because the length is not long enough. This explanation has been reemphasized in the revised paper, lines 402-406, and 428-432.
3) M(z) can be calculated directly by H(z), and the estimation of M(z) with LMS algorithm seems unnecessary.
Authors’ response:
M(z) includes the transfer function of interpolator, decimator, and two lowpass filters H(z). Although, it is easy to calculate directly decimated-band filter of one H(z), it is much more complicated to calculate decimated-band filter for cascade of two H(z). Besides, the transfer function of interpolator and decimator can’t be calculated directly. Thus, estimation by LMS algorithm is found suitable in this case. We have put the reason in the revised paper, shown in Section 2.2.1.
4) The manuscript in general is not well presented, with any grammatical and typo errors. A few are listed as follows.
(1) Line 27, “noises producing by rotating machines” should be “noises produced by rotating machines.”
(2) Line 239, “depend on” should be “depends on.”
(3) Line 254, “2.3.1” should be “2.3.2.”
(4) Some abbreviations appear without the corresponding explanations.
Authors’ response:
Thank you very much for your great help. The grammatical and typo errors have been revised carefully. Explanations about the abbreviations have been checked carefully.
Reviewer 2 Report
The submitted paper proposes multirate processing instead of single-rate in conventional audio-integrated feedback active noise control system. I would say that is quite an innovative study and it is correctly written. Very few comments are reported to the authors in order to improve the papers quality at best.
The beginning of chapter 2 is too similar to the last part of introduction. Please rephrase it.
Conclusions are too short. Please use this chapter to better summarize the work you did, how you reached the results, what are the limitations and the future perspectives.
References section is too short. I suggest the author to improve it by mentioning some more paper in the introduction. This will also help increasing the introduction itself. More references should reported on similar work, which means, the background information. Other could be added to enrich the field of application of the work. Thus, I suggest the authors to look at all possible fields where reducing noise would be beneficial. My only direct suggestion on the moment is soundscape improvement, as reducing inconvenient or unwanted noise has been demonstrated to positively act on human health (Cassina, Luca, et al. "Audio-visual preferences and tranquillity ratings in urban areas." Environments 5.1 (2018): 1; Thompson, Marie. Beyond Unwanted Sound: Noise, affect and aesthetic moralism. Bloomsbury Publishing USA, 2017; Nilsson, Mats E., et al. "Auditory masking of wanted and unwanted sounds in a city park." Noise Control Engineering Journal 58.5 (2010): 524-531.).
Author Response
Responses to the reviewers
We would like to thank the editor’s and reviewers’ comments, which are all considered in the revised version of the manuscript to improve its quality. The modifications are highlighted using color words corresponding to the reviewers in the revised manuscript. The responses are summarized as follows:
Reviewer 2:
1.The submitted paper proposes multirate processing instead of single-rate in conventional audio-integrated feedback active noise control system. I would say that is quite an innovative study and it is correctly written. Very few comments are reported to the authors in order to improve the papers quality at best.
The beginning of chapter 2 is too similar to the last part of introduction. Please rephrase it.
Authors’ response:
The beginning of section 2 has been rephrased.
2. Conclusions are too short. Please use this chapter to better summarize the work you did, how you reached the results, what are the limitations and the future perspectives.
Authors’ response:
The conclusions have been rewritten with additional parts, including the problem discussed in the paper, motivation of proposed system, difference of conventional and proposed system, how the results are reached, limitations and future work.
3. References section is too short. I suggest the author to improve it by mentioning some more paper in the introduction. This will also help increasing the introduction itself. More references should reported on similar work, which means, the background information. Other could be added to enrich the field of application of the work. Thus, I suggest the authors to look at all possible fields where reducing noise would be beneficial. My only direct suggestion on the moment is soundscape improvement, as reducing inconvenient or unwanted noise has been demonstrated to positively act on human health (Cassina, Luca, et al. "Audio-visual preferences and tranquility ratings in urban areas." Environments 5.1 (2018): 1; Thompson, Marie. Beyond Unwanted Sound: Noise, affect and aesthetic moralism. Bloomsbury Publishing USA, 2017; Nilsson, Mats E., et al. "Auditory masking of wanted and unwanted sounds in a city park." Noise Control Engineering Journal 58.5 (2010): 524-531.).
Authors’ response:
Thank you for the valuable suggested references. The references suggested by Reviewer have been cited on line 34, and included in References [6-8]. Besides, some more papers have also been cited to improve the introduction part, shown in lines 28-59, and included in References [1-5, 9-13, 16-19, 24].
Reviewer 3 Report
This paper describes a modified adaptive feedback ANC arrangement where a set of down samplers and up samplers are added to signal path, allowing the adaptive feedback ANC to run simultaneously with music playback, but at a lower sampling rate, so as to reduce computational cost of the ANC algorithm.
Overall the technique is sound, which is also backed up by simulation data. However, the reviewer has concern about the proposed ANC arrangement: Typically for feedback ANC structures, the best practice is to increase the sampling rate as much as possible, often much higher than 48kHz, so as to minimize the non-minimum phase delay introduced to the secondary channel by the down-sampling process. A longer delay in the secondary channel not only leads to slower convergence, but also limits the controllable bandwidth of the feedback ANC.
For a tonal noise such as those used in the authors' simulation, this would not have much of impact, as tonal noise is technically zero bandwidth and fully predictable. If the authors use a shaped wideband signal as noise source, significant performance drop may be observed when using the down-sampling scheme as proposed by the paper. This tradeoff should be simulated or at least be mentioned in the paper.
Another question is regarding the offline secondary channel estimation. What's the benefit of using this scheme, compared to simply playing a white noise at normal music sampling rate, and allowing the channel estimation to converge using the on-line estimation method?
The use of "subband" is a bit confusing, as it is often used in multi-band signal processing system. In this case, the "subband" is only referring to the decimated low sampling rate band. The reviewer suggest changing it to another term such as "low frequency band" or "decimated band".
Specific comments:
Line 256: "will has" should be "will have".
Figure 8: Suggest using logarithm x-axis for clearer data presentation.
Author Response
Responses to the reviewers
We would like to thank the editor’s and reviewers’ comments, which are all considered in the revised version of the manuscript to improve its quality. The modifications are highlighted using color words corresponding to the reviewers in the revised manuscript. The responses are summarized as follows:
Reviewer 3:
1.This paper describes a modified adaptive feedback ANC arrangement where a set of down samplers and up samplers are added to signal path, allowing the adaptive feedback ANC to run simultaneously with music playback, but at a lower sampling rate, so as to reduce computational cost of the ANC algorithm.
Overall the technique is sound, which is also backed up by simulation data. However, the reviewer has concern about the proposed ANC arrangement: Typically for feedback ANC structures, the best practice is to increase the sampling rate as much as possible, often much higher than 48kHz, so as to minimize the non-minimum phase delay introduced to the secondary channel by the down-sampling process. A longer delay in the secondary channel not only leads to slower convergence, but also limits the controllable bandwidth of the feedback ANC.
For a tonal noise such as those used in the authors' simulation, this would not have much of impact, as tonal noise is technically zero bandwidth and fully predictable. If the authors use a shaped wideband signal as noise source, significant performance drop may be observed when using the down-sampling scheme as proposed by the paper. This tradeoff should be simulated or at least be mentioned in the paper.
Authors’ response:
Thank you for your comments. It is true that the additional delay will limit controllable bandwidth. The purpose of the proposed work is to reduce narrowband noise while delivering wideband audio signal. The revised paper has clearly point out this tradeoff to make it more readable (lines 56-59, 264-266, and 511-513).
2.Another question is regarding the offline secondary channel estimation. What's the benefit of using this scheme, compared to simply playing a white noise at normal music sampling rate, and allowing the channel estimation to converge using the on-line estimation method?
Authors’ response:
To obtain good estimate of the secondary-path , the audio signal must be a persistent excitation and uncorrelated with the primary noise (has been addressed in lines 224-226). So, white noise signal satisfies this condition for on-line estimation. However, on-line estimation using white noise as excitation suffers from slow convergence problem, which has been addressed in lines 325-327. It is the reason why we suggest the offline method.
3.The use of "subband" is a bit confusing, as it is often used in multi-band signal processing system. In this case, the "subband" is only referring to the decimated low sampling rate band. The reviewer suggest changing it to another term such as "low frequency band" or "decimated band".
Authors’ response:
The term “decimated band” has been updated in the revised paper. Thank you for your great suggestion.
4.Specific comments:
>>Line 256: "will has" should be "will have".
Authors’ response:
Thank you very much for your great help. Grammatical error has been corrected.
>>Figure 8: Suggest using logarithm x-axis for clearer data presentation.
Authors’ response:
Thank you very much for your suggestion. Figure 8 has been updated with logarithm scale on x-axis.
Round 2
Reviewer 1 Report
Most of the technical issues have been addressed. Just a minor revision recommendation: The title should emphasis that the proposed method is for attenuating narrowband noise.
Author Response
Thank you for the Reviewer’s recommendation. The title of the revised paper has been emphasized w
Thank you for the Reviewer’s recommendation. The title of the revised paper has been emphasized with ‘Reducing Narrowband Noises’.

Reviewer 3 Report
The paper has improved from the revision. Just one comment on line 57: The authors state that the internal model feedback filter does not suffer from the waterbed effect. This is not true, as [15] has pointed out, the internal model scheme with a given feedback filter is equivalent to a regular feedback controller. It hence still suffers from the waterbed effect, somewhat in the form of prediction error. The adaptation essentially pushes residual noise around in the error signal spectrum, such that the total error energy is minimized.
Author Response
Thank you for the Reviewer’s comment. The statement ‘the internal model feedback filter does not suffer from the waterbed effect’ has been removed from the revised paper. In addition, the problem of waterbed effect has been readdressed in line 56-57.
